# A “Square Box”-Structured Triboelectric Nanogenerator for Road Transportation Monitoring

**DOI:** 10.3390/polym14132695

**Published:** 2022-06-30

**Authors:** Zhuo Chen, Hanyi Wu, Zhike Xia, Jian Zou, Shengji Wang, Peiyong Feng, Yuejun Liu, Zhi Zhang, Yinghui Shang, Xin Jing

**Affiliations:** 1Key Laboratory of Advanced Packaging Materials and Technology of Hunan Province, Hunan University of Technology, Zhuzhou 412007, China; cz100520@163.com (Z.C.); w18767766353@163.com (H.W.); xiazhike@126.com (Z.X.); zj980428@163.com (J.Z.); wangsj418@163.com (S.W.); fpyedu@163.com (P.F.); yjliu_2005@126.com (Y.L.); 2National & Local Joint Engineering Research Center for Advanced Packaging Material and Technology, Hunan University of Technology, Zhuzhou 412007, China; 3Shenzhen Weijian Wuyou Technology Co., Ltd., Shenzhen 518102, China; zhangzhiuse@163.com (Z.Z.); shyh1103@163.com (Y.S.)

**Keywords:** triboelectric nanogenerator, vibrational states, simulation analysis, road transportation

## Abstract

Nowadays, with the rapid development of e-commerce, the transportation of products has become more and more frequent. However, how to monitor the situation of products effectively and conveniently during road transportation is a long-standing problem. In order to meet this problem in practical applications, we fabricated a triboelectric nanogenerator sensor with a “square box” structure (S-TENG) for detecting the vibration suffered by vehicles. Specifically, with the spring installed in the S-TENG as a trigger, the two friction layers can contact and then separate to generate the real-time electrical signals when the S-TENG receives external excitation. The output voltage signals of the S-TENG under different vibration states were tested and the results demonstrated that the peak and zero positions of the open-circuit voltage–output curve are related to amplitude and frequency, respectively. In addition, the subsequent simulation results, obtained by ANSYS and COMSOL software, were highly consistent with the experimental results. Furthermore, we built a platform to simulate the scene of the car passing through speed bumps, and the difference in height and the number of speed bumps were significantly distinguished according to the output voltage signals. Therefore, the S-TENG has broad application prospects in road transportation.

## 1. Introduction

With the rapid development of the economy, numerous products are flowed by road transportation. However, the vehicles are vulnerable to vibration due to the uneven road surface, airflow disturbance, and vehicle engines, resulting in inevitable damage of the products during the process of road transportation. To avoid breakage, the products are always packaged with more cushioning materials. Therefore, over-packaging often occurs due to the lack of relevant data about the product during transportation as a reference. Hence, it is essential to monitor the vibration received by the products during transportation. Although acceleration sensors can detect vibration, they are not widely applied in road transportation due to their high price. Therefore, it is highly required to provide an additional and more economical way to monitor the vibration of products, which not only can provide predictive maintenance for the transportation vehicles but can also provide guidance for the proper packaging of the products.

Since it was first reported in 2012, the triboelectric nanogenerator (TENG) that can collect energy from the ambient mechanical energy has attracted widespread attention in the self-powered equipment field [1,2,3,4], owing to its unique advantages, such as simple design, excellent output performance, and high integration [5,6]. The TENG can be broadly divided into four types: the vertical contact–separation mode [7,8,9,10], the sliding mode [11,12], the single electrode mode [13,14,15], and the freestanding triboelectric-layer mode [16,17], and the working principle of the TENG is a coupling of the triboelectric effect and electrostatic induction [18]. More specifically, opposite and equivalent charges will be generated at the interface, owing to the charge transferring, when two frictional materials with opposite triboelectric polarities are in contact with each other [19,20,21]. Therefore, it can serve as not only an energy harvester but also a flexible self-powered sensor [22]. When the TENG received external incentives, the real-time signals would be generated that enabled it to act as a self-driving sensor in pressure sensing [23,24,25], vibration sensing [26,27,28], motion sensing [29,30,31], etc. For example, Jin et al. [32]. prepared a TENG sensor using aluminum foil and Polyethylene 2,5-furandicarboxylate (PEF) film as friction materials to detect finger movements and have stable output signals. Moreover, a smart insole that can recognize and monitor various gait types, such as standing, walking, and running, exhibiting a fast response time of less than 56 ms, was fabricated by embedding a TENG [33]. These applications and properties of a TENG fully demonstrate its enormous potential in the field of self-powered sensors.

In this work, a TENG with a “square box” structure (S-TENG) was prepared by 3D-printing technology, which belonged to the vertical contact–separation triboelectric nanogenerator and consisted of two parts: a square shell and a rectangular block. A spring attached to the rectangular block is equivalent to a trigger, leading to the anode oxidation (AO)-treated aluminum foil as the anode and the PTFE as the cathode coming into contact and then separating when the S-TENG is subjected to vibration. Modal exciters were used to make the S-TENG in different vibrational states and obtain different output signals. It is evident that the value of the peak open-circuit voltage is positively correlated with the vibration amplitude and the number of peak open-circuit voltage occurrences corresponding to the vibration frequency. In addition, ANSYS software and COMSOL Multiphysics software were used to simulate and analyze the relative displacement and the potential distribution between the anode and the cathode, respectively. In addition, an experimental platform to simulate the scene of the car passing through the speed bumps during road transportation, in which foamed strips with different heights were used to act as the speed bumps, and the developed TENG was attached onto the goods in the model car. It was found that the S-TENG was capable of accurately reflecting the vibration state of the model car passing through the foamed strips, which demonstrated that the S-TENG holds great potential to monitor the vibration of products in road transportation.

## 2. Materials and Methods

### 2.1. Preparation of S-TENG

The S-TENG was composed of a square shell with dimensions of 100 mm × 100 mm × 100 mm, and a rectangular block with dimensions of 50 mm × 50 mm × 30 mm. Aluminum foil (100 mm × 100 mm), which was treated with anode oxidization (AO) process following our previous study [34], was adhered to the inner surface of the square shell. A copper wire was used to connect the aluminum foil to the external circuit. Then, a small AO-treated aluminum foil (50 mm × 50 mm) and a spring (K = 60 N/mm, height = 40 mm) were adhered to the upper and lower surface of the rectangular block, respectively. After that, a PTFE film with dimensions of 50 mm × 50 mm was pasted on the small aluminum foil and connected to the external circuit in the same way. Finally, the bottom of the square shell and the spring were fixed on the same plane to form a “square box” structure. These existing sizes of the rectangular block were determined after comprehensive consideration of detection range and signal strength. The size of the S-TENG as well as the spring were optimized based on our preliminary study.

### 2.2. Performance Evaluation of S-TENG

Different frequencies and amplitudes were loaded on the S-TENG in the vertical direction by using a modal exciter (DFT9150, Chuanglian Corp., Nanjing, China). The output voltage signals were recorded by using an oscilloscope (Rigol, ZDS3034 Plus, Zhiyuan Corp., Guangzhou, China). The S-TENG were firstly operated for 1 min prior to data acquisition to ensure that a stable output state was obtained.

### 2.3. The Process of Simulation Analysis

ANSYS software was used to simulate and analyze the motion process of the S-TENG devices. The geometry and material properties of the virtual model were determined based on the physical devices. A motion function f(x)=Asin(2πωt) was used as the input function to mimic the movement of the S-TENG device under different vibration states.

A two-dimensional model was established in the COMSOL Multiphysics software to reflect the relative position between the PTFE film and the aluminum film during the contact–separation process. The steady-state potential distributions of the S-TENG in different states were simulated based on the electrostatic (ES) function to investigate the positive and negative electrostatic field distribution.

## 3. Results and Discussion

The structural configuration of the S-TENGs includes two parts: a square shell and a rectangular block, as shown in Figure 1a. The AO-treated aluminum foil, which acts as an anode material and electrode, was adhered to the inner surface of the square shell. More aluminum foil as an electrode was attached to the upper surface of the rectangular block. Then, a piece of PTFE film used as the cathode material was attached to it, resulting in a sandwich-like overall structure. The physical model of the S-TENG is composed of two parts as shown in Figure 1b. The S-TENG was a vertical contact–separation triboelectric nanogenerator, and its working principle was attributed to the coupling effect of contact electrification and electrostatic induction. Therefore, a spring was fixed on the lower surface of the rectangular block to adjust the distance between the PTFE film and the AO-treated aluminum foil which was changed when the spring bounced up and down. The diversity in polarity between anode and cathode materials and the roughness of their surfaces play a crucial role in output performance [35]. Therefore, according to the triboelectric sequences, aluminum foil and PTFE with significant polar differences were chosen as the anode and cathode materials, respectively. Furthermore, the roughness of the aluminum foil was increased by the AO treatment which has been widely used in previous studies [36,37], and as shown in Figure 1a, many ravines were present in the treated aluminum foil, which demonstrated the successful treatment of the aluminum foil.

Figure 1c illustrates the working principle of the S-TENG device, and the whole working process was divided into four stages. In the initial stage, there was no inductive potential difference between the aluminum foil and the PTFE film in the device being in the equilibrium state. In the upward stage, the S-TENG received external mechanical excitation, then the spring began to vibrate, driving the rectangular block to move upwards, which made the two friction-layer surfaces come into contact with each other, and an equal amount of positive and negative charges would be generated on the aluminum foil and the PTFE film, respectively. Immediately following, the rectangular block began to move downward because of the gravity and the potential difference would form in an open circuit due to the flow of electrons, which would gradually increase as the PTFE and aluminum foil gradually separated. The largest potential difference was achieved when the PTFE film moved to the lowest position. In the back state, the rectangular block was bounced back by the spring, and the open-circuit voltage decreased accordingly until it disappeared, which means the open-circuit voltage decreased to zero when the two friction layers were in full contact with each other. The whole cycle would be repeated as the device vibrated, and the S-TENG could be regarded as an electronic pump during the whole working process. The two electrodes were separated after exchanging electrons under electrostatic induction and vibration, and the inductive potential difference would be generated as a result. The current would be formed during the movement of electrons.

Based on the working principle as shown in Figure 1c, it is evident that the open-circuit voltage signal generated by the S-TENG is closely related to the vibration it receives. Therefore, the output signal of the S-TENG could be used to reflect the vibration information. The variation trend of the voltage when the S-TENG was under different external mechanical excitation was explored. In order to investigate the effect of vibration amplitude on the output signal, the S-TENG was fixed on the modal exciter and the frequency was set as 4 Hz for testing. Then, the amplitude was gradually increased from 10 mm to 30 mm, and an oscilloscope was used to record the output voltage signals. As shown in Figure 2a–c, it was found that, as the amplitude increased, the peak open-circuit voltage increased significantly. When the amplitude was 10 mm, a 5 V peak open-circuit voltage was obtained. When the amplitude was increased to 30 mm, the peak open-circuit voltage reached 19.8 V. On the one hand, the rectangular block would compress the spring to a greater degree as the amplitude increased, making the maximum distance between the AO-treated aluminum foil and the PTFE film increase so that the open-circuit voltage increased accordingly. To confirm this conclusion, a 3D model of the S-TENG device (Figure 2d) was established using ANSYS software, and the geometry, size, and mass of each component were modeled to match the developed S-TENG. Because S-TENG devices generally work in a simple harmonic motion, a motion function f(x)=Asin(2πωt), in which “A” represents the amplitude and is set as 10, 20, and 30, respectively, was used as the input function to mimic the movement of the S-TENG device. As shown in Figure 2e–g, it is obvious that the largest relative displacement between the two friction materials increased with the increased “A”. On the other hand, the increased voltage signal was ascribed to the increased contact pressure which leads to a more effective contact area. Therefore, when the relationship between the vibration amplitude and the peak value of the open-circuit voltage was built, it was easy to detect the magnitude of the external vibration loaded on the S-TENG by reading the voltage value.

In addition to the amplitude, the influence of the vibration frequency on the output signal of the S-TENG devices was also researched by vertically vibrating the S-TENGs with different frequencies of 2 Hz, 3 Hz, and 4 Hz, under a constant vibration amplitude of 30 mm. The results are shown in Figure 3a–c. It was found that, as the vibration frequency increased from 2 Hz to 4 Hz, there was no significant difference in the value of the peak open-circuit voltage which was almost maintained at 20 V. This might be attributed to the fact that the peak output voltage is mainly related to two factors: the density of the transfer charge, which is related to the material itself, and the largest relative displacement between the two friction layers after separation. In addition, when the frequency was changed, the amplitude was still kept the same, and the separation distance between the two friction layers was the same regardless of the frequency; therefore, the peak value of the output voltage was the same. 

Moreover, the number of peak open-circuit voltage occurrences corresponded exactly to the vibration frequency within the same testing time. For example, when the test time was 2 s and the frequency was 2 Hz, there were four peak open-circuit voltage signals, and when the frequency increased to 3 Hz, there were six peak open-circuit voltage signals. Based on the prepared simulation model, we obtained the results as shown in Figure 3d–f that were consistent with the test results, by changing the “f” from 2 Hz to 4 Hz in the motion function. Therefore, we can use S-TENGs to detect the vibration frequency based on the number of peak open-circuit voltage signal occurrences.

Meanwhile, the current signals of the S-TENG under different vibration states were also recorded using an electrochemical working station. Figure 4a–c show the output current of the S-TENGs when the vibration amplitude is 20 mm, and the vibration frequency is 4 Hz, 3 Hz, and 2 Hz, respectively. It can be seen from the results that the time required to collect four peak signals corresponds to the vibration frequency. For example, when the frequency was 2 Hz, there were four peak current signals within 2 s. Moreover, the output current signal of the S-TENGs under different vibration amplitudes was also recorded, and the results show that the peak current increases from 5.0 × 10^−7^ to 2.0 × 10^−6^ with the vibration amplitude increasing from 10 mm to 30 mm as shown in Figure 4d–f.

In addition, COMSOL Multiphysics software was used to simulate the potential field distribution of the S-TENGs during the vibration process. To facilitate the analysis calculation, we established a two-dimensional equivalent model to obtain the relationship of the potential difference and the distance between the AO-treated aluminum foil and the PTFE film during their separation after contacting. As shown in Figure 5, two mutually parallel rectangles represent the anode and cathode of the S-TENG, respectively. The smaller potential difference at the initial stage during the separation process of the two electrodes can be observed, as shown in Figure 5a. As the distance gradually increased, the color difference became more significant (Figure 5b), which means the potential difference between the two electrodes gradually increased. The potential difference reached the maximum value when the distance reached the limit value, as shown in Figure 5c. Then, the cathode would move in the opposite direction, leading to a decrease in the distance between the two electrodes so that the potential difference gradually decreased and disappeared until the next contact, as shown in Figure 5d. The simulation results and the test result revealed the same regulation, in which the voltage signal was positively associated with the distance between the two electrodes.

An experimental platform was built up to simulate the application scenario in which the S-TENGs were applied in road transportation to detect the vibration encountered while passing over the speed bumps. The S-TENGs were installed on the model car, and a direct-current (DC) reciprocating motor was used to push the model car forward. Meanwhile, the foamed strips as speed bumps were placed on the route which the model car would pass through. The schematic and physical diagrams are shown in Figure 6a,b, respectively. The output voltage signal of the S-TENG is shown in Figure 6c when the model car passes through one foamed strip. Two peak open-circuit voltage signals can be attributed to the two vibrations caused when the front and rear wheels of the model car pass through the foamed strip, leading to two contact–separation processes between the two electrodes of the S-TENG. Furthermore, we placed two foamed strips with different heights on the route, and the output voltage signal is shown in Figure 6d. It can be clearly seen that there are two different peak voltages. The higher one is generated when the car passes through the higher foamed strips, and in contrast, a lower output signal was generated as the car passes through a lower foamed strip. Therefore, the amplitude and frequency information about vibrations could be successfully obtained via analyzing the voltage signal of the S-TENG, which indicates the great potential of the S-TENGs to be applied as vibration detectors in road transportation.

## 4. Conclusions

In conclusion, we successfully fabricated an S-TENG sensor with a “square box” structure composed of two homemade 3D printed molds: a square shell and a rectangular block. The S-TENG was fixed on the modal exciter and the effects of amplitude and frequency on the output voltage signal were explored by adjusting the parameters to put the S-TENG in different vibrational states. When the vibration amplitude increased from 10 mm to 30 mm, the peak open-circuit voltage accordingly increased from 5 V to 19.8 V. Meanwhile, when the vibration frequency increased from 2 Hz to 4 Hz with the same vibration amplitude, the peak number of the voltage signals in the unit time was the same as the vibration frequency. Moreover, the simulation analysis results, including the relative displacement and the potential distribution between the two friction layers of the S-TENG, sufficiently corroborated these actual test results. Furthermore, a vibration monitoring platform was built to simulate the scenario of the vehicle passing through the speed bumps using the S-TENG as sensors. The test results were consistent with our assumption that the number of voltage peak signals was twice the number of the speed bumps that the vehicle passed through, and the higher the speed bump, the larger the voltage, which indicated that the developed S-TENG paves a way for the application of triboelectric nanogenerators in road transportation.

## Figures and Tables

**Figure 1 polymers-14-02695-f001:**
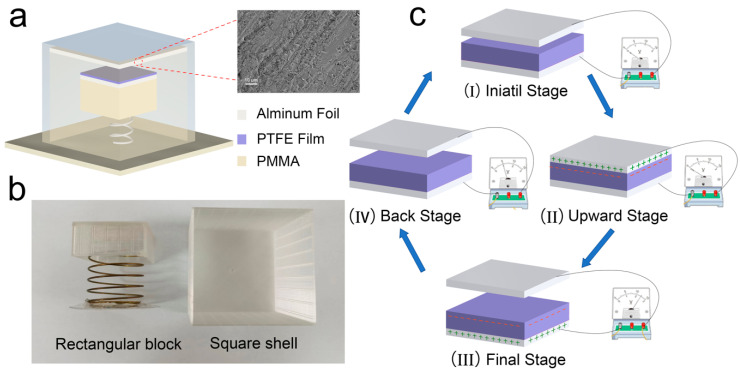
(**a**) Schematic illustration of the S-TENG device. (**b**) Digital photographs of the two components of S-TENG. (**c**) Schematic diagram of the working principles of the developed S-TENG.

**Figure 2 polymers-14-02695-f002:**
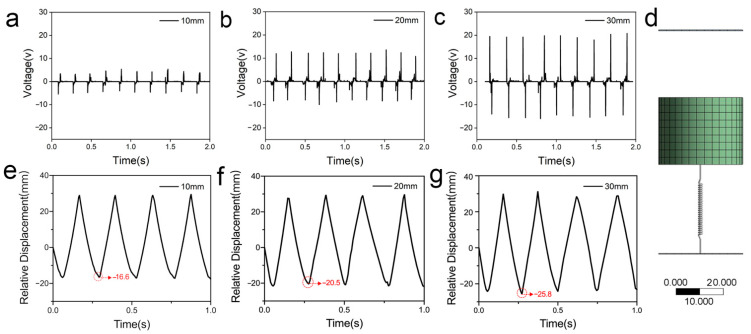
Output voltage signals and relative displacement at different vibration amplitude from 10 mm to 30 mm and same frequency of 4 Hz. (**a**) Output voltage signal when the vibration amplitude is 10 mm. (**b**) Output voltage signal when the vibration amplitude is 20 mm. (**c**) Output voltage signal when the vibration amplitude is 30 mm. (**d**) Simulation model of S-TENG. (**e**) Relative displacement when the vibration amplitude is 10 mm. (**f**) Relative displacement when the vibration amplitude is 20 mm. (**g**) Relative displacement when the vibration amplitude is 30 mm.

**Figure 3 polymers-14-02695-f003:**
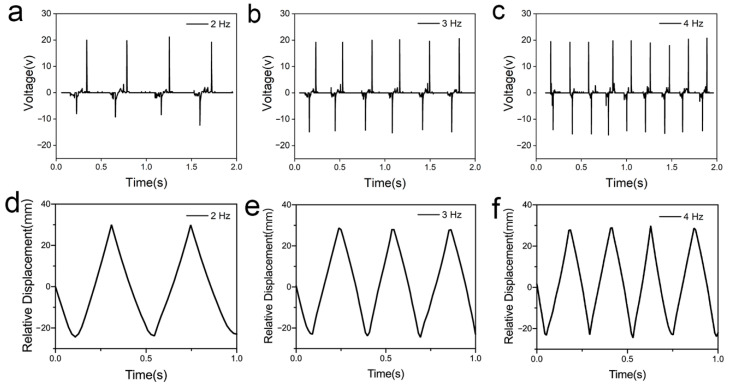
Output voltage signals and relative displacement at different vibration frequency from 2 Hz to 4 Hz and same amplitude of 30 mm. (**a**) Output voltage signal when the vibration frequency is 2 Hz. (**b**) Output voltage signal when the vibration frequency is 3 Hz. (**c**) Output voltage signal when the vibration frequency is 4 Hz. (**d**) Relative displacement when the vibration frequency is 2 Hz. (**e**) Relative displacement when the vibration frequency is 3 Hz. (**f**) Relative displacement when the vibration frequency is 4 Hz.

**Figure 4 polymers-14-02695-f004:**
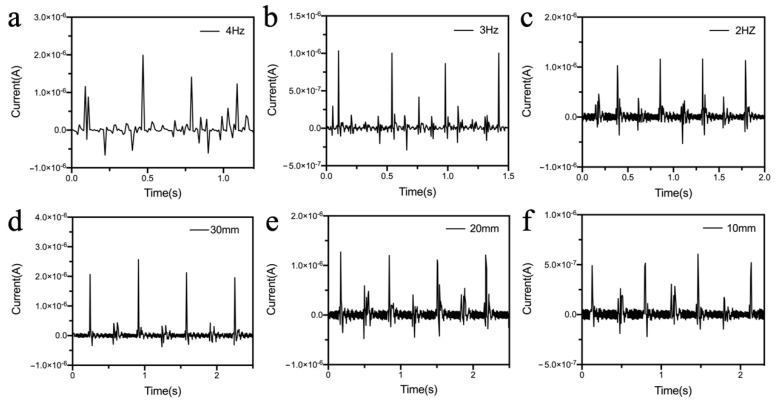
Output current signals at different vibration statue. (**a**) The vibration frequency is 4 Hz and the vibration amplitude is 20 mm. (**b**) The vibration frequency is 3 Hz and the vibration amplitude is 20 mm. (**c**) The vibration frequency is 2 Hz and the vibration amplitude is 20 mm. (**d**) The vibration frequency is 2 Hz and the vibration amplitude is 30 mm. (**e**) The vibration frequency is 2 Hz and the vibration amplitude is 20 mm. (**f**) The vibration frequency is 2 Hz and the vibration amplitude is 10 mm.

**Figure 5 polymers-14-02695-f005:**
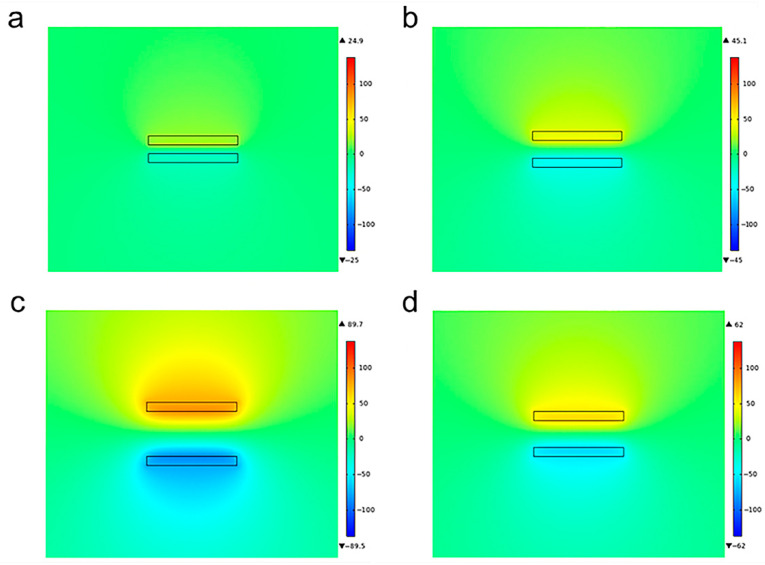
COMSOL Multiphysics simulation results for the S-TENGs at different states. (**a**) Early stage after separation. (**b**) Middle stage after separation. (**c**) Final stage of separation. (**d**) Recovery stage.

**Figure 6 polymers-14-02695-f006:**
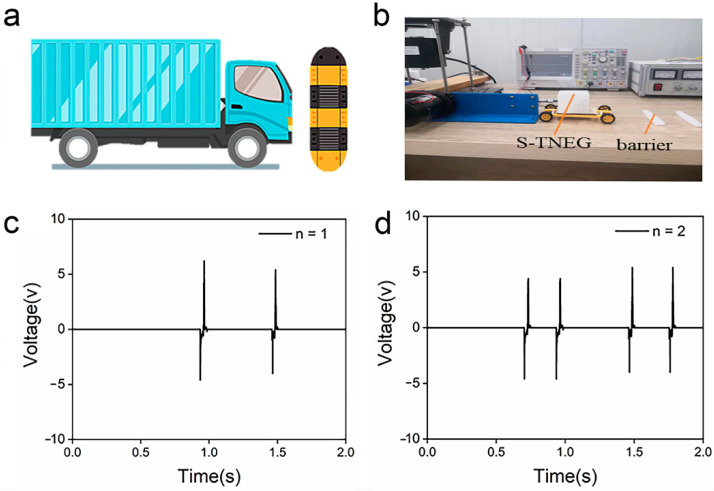
(**a**) Schematic diagram of the simulation platform. (**b**) Digital photographs of the simulation platform. (**c**) Test result when there was only a single speed bump. (**d**) Test result when there were two speed bumps.

## Data Availability

Not applicable.

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
