# Peer review of "A “Square Box”-Structured Triboelectric Nanogenerator for Road Transportation Monitoring"

_polymers, 2022, doi:10.3390/polym14132695_

Round 1
Reviewer 1 Report
Dear Authors, in spite of the fact that I consider that the overall merit of the article is high, and it has a relevant interest and scientific soundness. There are some questions that I would want to be answer and some changes that I think are necessary to recommend this article for publication.
1) The authors have proposed a triboelectric sensor for detecting the vibration suffered by vehicles, I don’t find the relationship between this, and the over packaged and the reduction of waste, that is included in the introduction section. This aspect needs to be clarified. Maybe the predictive maintenance of road transportation vehicles is a more logical application for the triboelectric generator.
2) Section 2 must be reorganized, inside this section there are a lot small subsections which don’t give so much information. Maybe it is not necessary to add more information, only reorganize it, you can complete these subsections with some information from the results section, because this section is so long and all the information do not correspond to results.
3) The conclusion section must be improved. There are sufficient results in this research to draw a more detailed conclusion.
Author Response
Reply to the reviewer comments on the paper entitled “(polymers-1738543) A “Square Box” Structured Triboelectric Nanogenerator for Road Transportation Monitoring”
We are deeply grateful for the reviewer’ suggestions and comments on this paper. They have helped to improve the quality of our paper and the current research. We have made the following changes to accommodate their comments and suggestions in the revision.
Reviewers’ comments:
Reviewer #1:
Dear Authors, in spite of the fact that I consider that the overall merit of the article is high, and it has a relevant interest and scientific soundness. There are some questions that I would want to be answer and some changes that I think are necessary to recommend this article for publication.
- The authors have proposed a triboelectric sensor for detecting the vibration suffered by vehicles, I don’t find the relationship between this, and the over packaged and the reduction of waste, that is included in the introduction section. This aspect needs to be clarified. Maybe the predictive maintenance of road transportation vehicles is a more logical application for the triboelectric generator.
Reply to the comment:
Thanks for the suggestion. It was a great idea to use S-TENG for predictive maintenance of road transportation vehicles, providing a new direction for our future research. We are sorry to make the confusion about the relationship between the overpackaging and monitoring the vibration state of a product in transportation.
Here is my interpretation of the problem. During the road transportation, it is inevitable for the vehicles to endure irregular vibration due to the large bumps on the road, which make the loaded products suffer from large vibration, resulting in damage to the products. Therefore, researchers also provide packaging solutions based on their experience to protect the products which always result in excessive packaging due to the lack of relevant information about products during transportation as a reference. Therefore, we hope to use S-TENG to collect the vibration information of products during transportation, which can provide the guideline for the design of packaging solutions to reduce material waste.
Furthermore, we have also made corresponding changes in the Introduction in the revised manuscript as follows: “With the rapid development of the economy, numerous products are flowed by road transportation. However, the vehicles are vulnerable to vibration due to the uneven road surface, airflow disturbance and vehicle engines, resulting in inevitable damage for the products during the process of road transportation. To avoid breakage, the products are always packaged with more cushioning materials. Therefore, over-packaging often occurs due to the lack of relevant data about the product during transportation as a reference. Hence, it is essential to monitor the vibration that received by the products during transportation. Although acceleration sensors can detect vibration, they are not widely applied in road transportation due to their high price. Therefore, it is highly required to provide an additional and more economical way to monitor the vibration of products, which not only can provide predictive maintenance for the transportation vehicles, but also can provide guidance for the proper packaging of the products.”
- Section 2 must be reorganized, inside this section there are a lot small subsections which don’t give so much information. Maybe it is not necessary to add more information, only reorganize it, you can complete these subsections with some information from the results section, because this section is so long and all the information do not correspond to results.
Reply to the comment:
Thanks for the suggestion. According to the suggestion, we have simplified the Section 2 as follows:
2.1. Preparation of S-TENG
The S-TENG was composed of a square shell with dimensions of 100 mm ´ 100 mm ´ 100 mm, and a rectangular block with dimensions of 50 mm ´ 50 mm ´ 30 mm. An aluminum foil (100 mm ´ 100 mm) which was treated with anode oxidization (AO) process following our previous study [34] was adhered to the inner surface of the square shell. A copper wire was used to connect the aluminum foil to the external circuit. Then, a small AO-treated aluminum foil (50 mm ´ 50 mm) and a spring (K=60 N/mm,height=40 mm) were adhered to the upper and lower surface of the rectangular block, respectively. After that, a PTFE film with dimensions of 50 mm ´ 50 mm was pasted on the small aluminum foil, and connected to the external circuit in the same way. Finally, the bottom of the square shell and the spring were fixed on the same plane to form a “square box” structure. These existing sizes of the rectangular block was determined after comprehensive consideration of detection range and signal strength. The size of the S-TENG as well as the spring were optimized based on our preliminary study.
2.3. The Process of Simulation Analysis
ANSYS software was used to simulate and analyze the motion process of the S-TENG devices. The geometry and material properties of the virtual model were determined based on the physical devices. A motion function was used as the input function to mimic the movement of the S-TENG device under different vibration state.
A two-dimensional model was established in the COMSOL Multiphysics software to reflect the relative position between the PTFE film and the aluminum film during the contact separation process. The steady-state potential distributions of the S-TENG in different states were simulated based on the electrostatic (ES) function to investigate the positive and negative electrostatic field distribution.
- The conclusion section must be improved. There are sufficient results in this research to draw a more detailed conclusion.
Reply to the comment:
Thanks for the suggestion. According to the suggestion, we have drawn a more detailed conclusion in the revised manuscript based on the results as follows: “In conclusion, we successfully fabricated an S-TENG sensor with a “square box” structure composed of two homemade 3D printed molds: a square shell and a rectangular block. The S-TENG was fixed on the modal exciter and the effects of amplitude and frequency on the output voltage signal were explored by adjusting the parameters to make the S-TENG in different vibrational states. When the vibration amplitude increased from 10 mm to 30 mm, the peak open-circuit voltage accordingly increased from 5 V to 19.8 V. Meanwhile, when the vibration frequency increased from 2 Hz to 4 Hz with the same vibration amplitude, the peak number of the voltage signals in the unit time was the same as the vibration frequency. Moreover, the simulation analysis results, including the relative displacement and the potential distribution between the two friction layers of the S-TENG, sufficiently corroborated these actual test results. Furthermore, a vibration monitoring platform was built up to simulate the scenario of the vehicle passing through the speed bumps using the S-TENG as sensors. The test results were consistent with our assumption that the number of voltage peak signals was twice the number of the speed bumps that the vehicle passed through, and the higher the speed bump, the larger the voltage, which indicated that the developed S-TENG paves a way for the application of triboelectric nanogenerators in the road transportation.”

Reviewer 2 Report
1. Similar idea had been proposed in literature (Nano Energy 78 (2020) 10517 and Nano Energy 97 (2022) 107219). Please emphasize the novel part of this work.
2. According to figure 1, S-TENG vertically connects the rectangular block in the system, the effect of different weights may impact on the TENG output signal. Please explain more detail about how to avoid inference effect in this system to realize the authentic output.
3. Figures 2 and 3 show the output voltage with different conditions. In general, the TENG output was reported both in voltage and current. Therefore, it would be better to include the output current.
Author Response
eviewer #2:
- Similar idea had been proposed in literature (Nano Energy 78 (2020) 10517 and Nano Energy 97 (2022) 107219). Please emphasize the novel part of this work.
Reply to the comment:
Thanks for the comment. The suggested paper, (Nano Energy 97 (2022) 107219) is a review article in which the materials, structure designs and corresponding performance of TENG for cooperative vehicle infrastructure system (CVIS) among human (pedestrian and driver), vehicle (automobile, non-motorized vehicle and intelligent tire), road and environment were summarized to promote the triboelectric self-powered sensing application to intelligent transportation systems (ITS) fields, which indicated that the TENG have great potential in many fields due to its functions of energy harvesting and sensing detection. For the other suggested paper (Nano Energy 78 (2020) 10517), we could not find it, which might be due to the wrong page information. We are truly sorry for that.
Compared to the previous studies, the novelty of our work is as follows: we specifically designed a square triboelectric nanogenerator (S-TENG) with springs as triggers based on the principle of contact-separated triboelectric nanogenerators and application scenarios, which was proposed to monitor the real-time vibration state of products during transportation. Moreover, which also could provide some guidance for the rational transportation packaging. In addition, we further verified the reliability of S-TENG in this application field by the simulation analysis software.
- According to figure 1, S-TENG vertically connects the rectangular block in the system, the effect of different weights may impact on the TENG output signal. Please explain more detail about how to avoid inference effect in this system to realize the authentic output.
Reply to the comment:
Thanks for the valuable suggestion. Based on our preliminary study for the S-TENG, we also found that the weight of the rectangular block indeed affected the output signal of the TENG. When the weight of the rectangular block was larger, it will prevent the spring from driving the PTFE film to move upward so that it cannot contact the aluminum foil. Therefore, the S-TENG would not output an electrical signal under slight vibration, that is, increasing the weight would improve the minimum detection limit of the S-TENG. In contrast, when the mass of the rectangular block was smaller, the distance of the spring travelling down was reduced, resulting in a weaker output signal. Therefore, the size of the S-TENG was determined based on the preliminary study. To avoid confusion, we made the following change in the Section 2 in the revised manuscript as follows: “The size of the S-TENG as well as the spring were optimized based on our preliminary study.”
- Figures 2 and 3 show the output voltage with different conditions. In general, the TENG output was reported both in voltage and current. Therefore, it would be better to include the output current.
Reply to the comment:
Thanks for the valuable suggestion. We also tried to collect the current signal during the experiments. When the amplitude was 30 mm and the frequency was 4 Hz, the current signal collected is shown in Figure 1. It can be found that the noise signal is too obvious, resulting in the disordered current signal, which was caused by the low voltage signal and large resistance of the setup. Therefore, we did not present the current signal in the manuscript.
